# The impact of self-isolation on psychological wellbeing in adults and how to reduce it: A systematic review

**Alex F. Martin**[1,2]*, **Louise E. Smith**[1,2], **Samantha K. Brooks**[1,2], **Madeline V. Stein**[1],
**Rachel Davies**[1], **Richard Amlôt**[2,3], **Neil Greenberg**[1,2], **Gideon James Rubin**[1,2]

**1** Institute of Psychiatry, Psychology and Neuroscience, King's College London, London, United Kingdom,
**2** NIHR Health Protection Research Unit in Emergency Preparedness and Response, London, United
Kingdom, **3** Chief Scientific Officer's Group, United Kingdom Health Security Agency, London, United
Kingdom

* alex.f.martin@kcl.ac.uk

Multicultural Health and Community Services:
Access Alliance, CANADA

**Peer Review History:** PLOS recognizes the
benefits of transparency in the peer review
process; therefore, we enable the publication
of all of the content of peer review and
author responses alongside final, published
articles. The editorial history of this article is
available here: https://doi.org/10.1371/journal.
pone.0310851

## Abstract

### Objective

To synthesise evidence on the impact of self-isolation at home on the psychological and
emotional wellbeing of adults in the general population during the COVID-19 pandemic.

### Methods

This systematic review was registered on PROSPERO (CRD42022378140). We searched
Medline, PsycINFO, Web of Science, Embase, and grey literature. Wellbeing included adverse
mental health outcomes and adaptive wellbeing. We followed PRISMA and synthesis without
meta-analysis (SWiM) guidelines. We extracted data on the impact of self-isolation on wellbe-
ing, and factors associated with and interventions targeting wellbeing during self-isolation.

### Results

Thirty-six studies were included. The mode quality rating was 'high-risk'. Depressive and
anxiety symptoms were most investigated. Evidence for an impact of self-isolation on
wellbeing was often inconsistent in quantitative studies, although qualitative studies con-
sistently reported a negative impact. People with pre-existing mental and physical health
needs reported increased symptoms of mental ill health during self-isolation. Studies
reported modifiable stressors that have been reported in previous infectious disease
contexts, such as inadequate support, poor coping strategies, inadequate and conflicting
information, and highlighted the importance of regular contact from trusted healthcare
professionals. Interventions targeting psychological wellbeing were rare and evaluative
studies of these had high or very high risk of bias.

### Conclusion

When implementing self-isolation directives, public health officials should prioritise support
for individuals who have pre-existing mental or physical health needs, lack support, or

**Data availability statement:** All relevant data are within the manuscript and its Supporting Information files.

**Funding:** This study was funded by the Research England Policy Support Fund 2022-23 (from the allocation to King's College London). The funders had no role in the study design, the collection, analysis and interpretation of data, in the writing of the article, or in the decision to submit it for publication. AFM, LES, SKB, RA and GJR are supported by the National Institute for Health and Care Research Health Protection Research Unit (NIHR HPRU) in Emergency Preparedness and Response, a partnership between the UK Health Security Agency, King's College London and the University of East Anglia. The views expressed are those of the authors and not necessarily those of the NIHR, UKHSA or the Department of Health and Social Care. For the purpose of open access, the author has applied a Creative Commons Attribution (CC BY) licence to any Author Accepted Manuscript version arising.

**Competing interests:** This work was carried out at King's College London. LES, RA, and GJR were participants of the UK's Scientific Advisory Group for Emergencies or its subgroups. GJR advised the UK's Office for National Statistics on its work relating self-isolation – papers relating to this work were considered as part of the review. All authors co-authored papers that were considered during the review process. RA is an employee of the UK Health Security Agency. AFM, SKB, RD, MVS, and NG report no competing interests. This does not alter our adherence to PLOS ONE policies on sharing data and materials.

who are facing significant life stressors. Focus should be directed toward interventions that address loneliness, worries, and misinformation, whilst monitoring and identifying individuals in need of additional support.

## Introduction

Self-isolation is a critical strategy in global efforts to curb infectious diseases. Self-isolation in this study is defined as both isolation (separating those who are sick from those who are well) and quarantine (separating those at risk of illness from those who are well) [1].

Most studies during the COVID-19 pandemic focused on the impact of "lockdown" measures (broad population stay-at-home orders) on psychological and emotional wellbeing (hereafter *wellbeing*) [2]. However, the impact of home-based self-isolation on adult wellbeing globally has not yet been systematically reviewed.

This is important given that a) studies carried out in other infectious disease contexts indicate that self-isolation may be associated with psychopathology symptoms and broader wellbeing outcomes in adults, such as insomnia and substance use [3,4], b) self-isolation has the potential to impact specific aspects of mental health, such as social stigma and anxiety related to infection or prognosis [3–6], and c) it is likely that home-based self-isolation will be used in future outbreaks of infectious disease, as with mpox in 2022 [7]. Reducing the burden of self-isolation on those affected remains a priority for public health and clinical practice.

This systematic review appraises:

1. The impact of self-isolation during the COVID-19 pandemic on wellbeing.

2. Factors associated with wellbeing outcomes during or following self-isolation.

3. The effectiveness of interventions designed to improve wellbeing during or following self-isolation.

## Methods

This systematic literature review was carried out in accordance with the Cochrane Collaboration guidelines for the conduct of systematic reviews [8], and the Preferred Reporting Items for Systematic Reviews and Meta-Analyses (PRISMA; see S1 Appendix) [9,10]. The protocol was prospectively registered on PROSPERO (CRD42022378140). The biggest change from the protocol was to only include studies of adults in this review, due to the number of studies identified in the initial search. Other deviations from the protocol are reported in full in S2 Appendix. As we performed a systematic review of previously published literature, ethical approval and participant consent were not required. This is because our research did not involve interaction with participants or the use of individual participant data.

A systematic search was conducted of studies published between 01 January 2020 and 13 December 2022. The peer-review and preprint searches were carried out between 14-17 December 2022, backward reference checking continued until 16 February 2023, and grey literature searches were conducted until 24 July 2023. We searched six peer-review and preprint databases (Medline, PsycInfo, Web of Science, Embase, PsyArXiv, medRxiv). In addition, we searched five grey literature databases (OpenGrey, World Health Organization [WHO], National Technical Information Service [NTIS; United States Department of Commerce], WorldCat, and Agency for Healthcare Research and Quality [AHRQ]), recommended by the National Institutes of Health Library (Literature Search: Databases and Gray Literature. National Institutes of Health Library). A Google search and searches of the websites for

relevant UK agencies and organisations (e.g., the UK Testing Initiatives Evaluation Board, the UK Office for National Statistics), and direct inquiries with UK Government agencies were used to identify other potentially relevant sources. We searched only UK agencies due to our team's experience. Combined with the UK-based Google search, this may have resulted in missing some grey literature from outside the UK (although we note that the searches we undertook did not find any studies for inclusion). A full description of the grey literature, organisation and Google searches, as well as a database search example are reported in the S3 Appendix.

The search strategy included terms for COVID-19 AND isolation and quarantine (combined with NOT social isolation) AND psychological wellbeing. Social isolation was excluded as a self-isolation search term because it generated a large number of citations that did not meet our definition of isolation or quarantine. Broad and specific search terms were used to maximise the detection of eligible studies. The search was also used as the basis of a separate systematic review exploring adherence to self-isolation [11], screening for both reviews up to full text stage was performed in parallel.

The search functions on the pre-print registers are not suitable for use in systematic reviews due to a number of limitations (specifically, confusing Boolean operators, a lack of reproducibility, and no batch export). A discussion and code are provided in this blog: https://ropensci.org/blog/2020/10/20/searching-medrxivr-and-biorxiv-preprint-data

To overcome this, preprint searches were extracted using the R package medrxiv [12]. The code is available on request from the corresponding author.

The search was piloted, and the reviewing team (AFM, LES, SKB, MVS, RD, and GJR) reviewed a training set of 300 studies. Discrepancies were discussed until agreement on included studies was attained. Piloting led to some revisions and clarifications of the protocol (see S2 Appendix). Then, reviewers independently screened citations, meeting weekly to reach agreement on queries and discrepancies.

Studies were included if they used original data to investigate the impact of self-isolation on wellbeing during the COVID-19 pandemic. Self-isolation was defined as: anyone advised (directly or by widely disseminated public health guidance) to avoid contact with others because they were known or suspected to have COVID-19 or because they were suspected to be incubating COVID-19. Wellbeing included adverse mental health outcomes and adaptive characteristics. We included adults who self-isolated at home and excluded children, healthcare workers, and those in managed isolation facilities or in a hospital. For aim 1, quantitative studies had to use a design that allowed attribution of the impact of self-isolation on wellbeing, for example, through use of a comparison to a control group. For aim 2, factors associated with wellbeing had to be directly related to, or occur during, the self-isolation period. For example, studies investigating the impact of a change in the national containment rules after the isolation period but before the study was carried out were excluded. For this aim, studies were included that compared home to isolation in a managed facility. Grey literature was only included if it investigated the effectiveness of an intervention (aim 3), to ensure only the most rigorous non-peer-reviewed studies were included and because of the dearth of peer-reviewed data on this specific topic. If it was unclear whether a study met the inclusion criteria, the corresponding author was contacted, and the study was excluded if no response was received.

Data were extracted by one reviewer (AFM for quantitative studies and SKB for qualitative studies) using a piloted, standardised table. All studies were discussed with at least one other reviewer (AFM, LES or GJR). Extracted data included: study characteristics (design, methods, sampling, demographics); isolation characteristics (reason, duration, context); and wellbeing characteristics (measures, impact, associated factors, interventions). We reported the most

rigorous analysis conducted in each study, for example, multivariable analysis over unadjusted analysis.

Study quality assessment was performed by one reviewer (AFM for quantitative studies and SKB for qualitative studies), all studies were discussed with at least one other reviewer (AFM, LES or GJR). For quantitative studies, we used the Risk of Bias in Non-randomized Studies for Exposure (ROBINS-E) and Interventions (ROBINS-I) [13,14], guided by the *Cochrane Handbook for Systematic Reviews of Interventions* for assessing risk of bias of different types of non-randomized studies [8]. ROBINS-E and ROBINS-I require the pre-identification of potentially significant confounding domains. We specified gender and age, which were applied to all studies. Using the ROBINS algorithm, studies that do not achieve this a priori confounding criterion are not assessed further, due to their substantial risk of bias. Neverthe-less, we evaluated all risk of bias domains for all included studies to provide a comprehensive perspective on bias risk within each domain across the entire set of studies. In cases where studies did not meet the a priori confounding criterion, they were assigned an overall bias risk rating of 'very high.' ROBINS assessments are specific to a reported result rather than a study. Consequently, studies that reported a result for aim 1 and 2 received two risk of bias scores. Each result was categorised as low risk, some concerns, high risk, or very high risk based on the tool's algorithm. For consistency of nomenclature, ratings for the ROBINS-I were converted to use the same terminology as the ROBINS-E: ROBINS-I "low risk of bias" = low risk of bias; ROBINS-I "moderate risk of bias" = some concerns; ROBINS-I "serious risk of bias" = high risk of bias; ROBINS-I "critical risk of bias" = very high risk of bias). No amendments were made to the algorithms used to determine the risk of bias rating.

We used the Critical Appraisal Skills Programme (CASP) checklist for qualitative studies [15], but reworded the item 'how valuable is the research?' to 'do the authors discuss the value of the research in terms of implications and contribution to literature?' to allow yes/no responses in line with the other items and to give each study an overall quality score percentage. Scores were out of ten and reported as a percentage, with higher scores indicating better quality.

Quantitative data were synthesised narratively, following SWiM guidelines [16]. No meta-analysis was planned due to expected heterogeneity in study design, outcomes, and associated factors. Studies were synthesised within each research aim separately and grouped by psychological wellbeing outcome to reduce heterogeneity. Studies reporting general psychological symptom scores (i.e., not disorder-specific) or subjectively reported mental health (e.g., 'compared to before the COVID-19 pandemic, how would you say your mental health is now?' [17]) were grouped together as *general psychological symptoms*. To be included in the synthesis, each wellbeing outcome (aim 1) and each factor (aim 2) had to be investigated in at least two studies. If removing one study from the synthesised table for this reason left only one study for another outcome/factor, that study was also removed from the table.

For aim 1, we compared outcomes for those in self-isolation with those not self-isolating. For aim 2, we also grouped studies by factor (isolation, demographic, COVID-19, and mental/physical health characteristics) and associations between factors and self-isolation were reported. These groupings were not defined a priori, but the themes emerged during the process of extracting the factors and were revised iteratively as we synthesised the findings. For aim 3, we grouped studies by intervention type and compared pre- and post-intervention scores in the intervention group when there was no control group, and compared the intervention group to the control group where one was reported. Studies were synthesised using tabulation and vote counting based on the direction of effect. The number of studies, the consistency of effects across studies, and the risk of bias across studies were used to assess the certainty of synthesised findings.

Qualitative data were synthesised using meta-ethnography, following eMERGe guidelines [18]. Subthemes were developed by psychological wellbeing outcome and, for aim 2, subthemes were developed by factor. No qualitative studies explored interventions (aim 3). Qualitative findings were synthesised using reciprocal translation (to understand one study's findings in terms of another) and refutational translation (to explore differences between studies and differences identified in quantitative findings).

## Results

The search identified 15,275 unique citations (Fig 1). Following title and abstract screening, 987 studies underwent full-text screening (reported in S4 Appendix). Thirty-six studies were included, all of which were identified through database searches [17,19–53].

### Study characteristics

Sample sizes ranged from 14 to 18,146. One study included only older adults [22], all other studies included adults aged 18 years and over. Studies investigated the general population, rather than sub-groups within the population. Study characteristics (region, design, sample size, isolation reason, and lockdown context) are synthesised in Table 1 and full tables of the

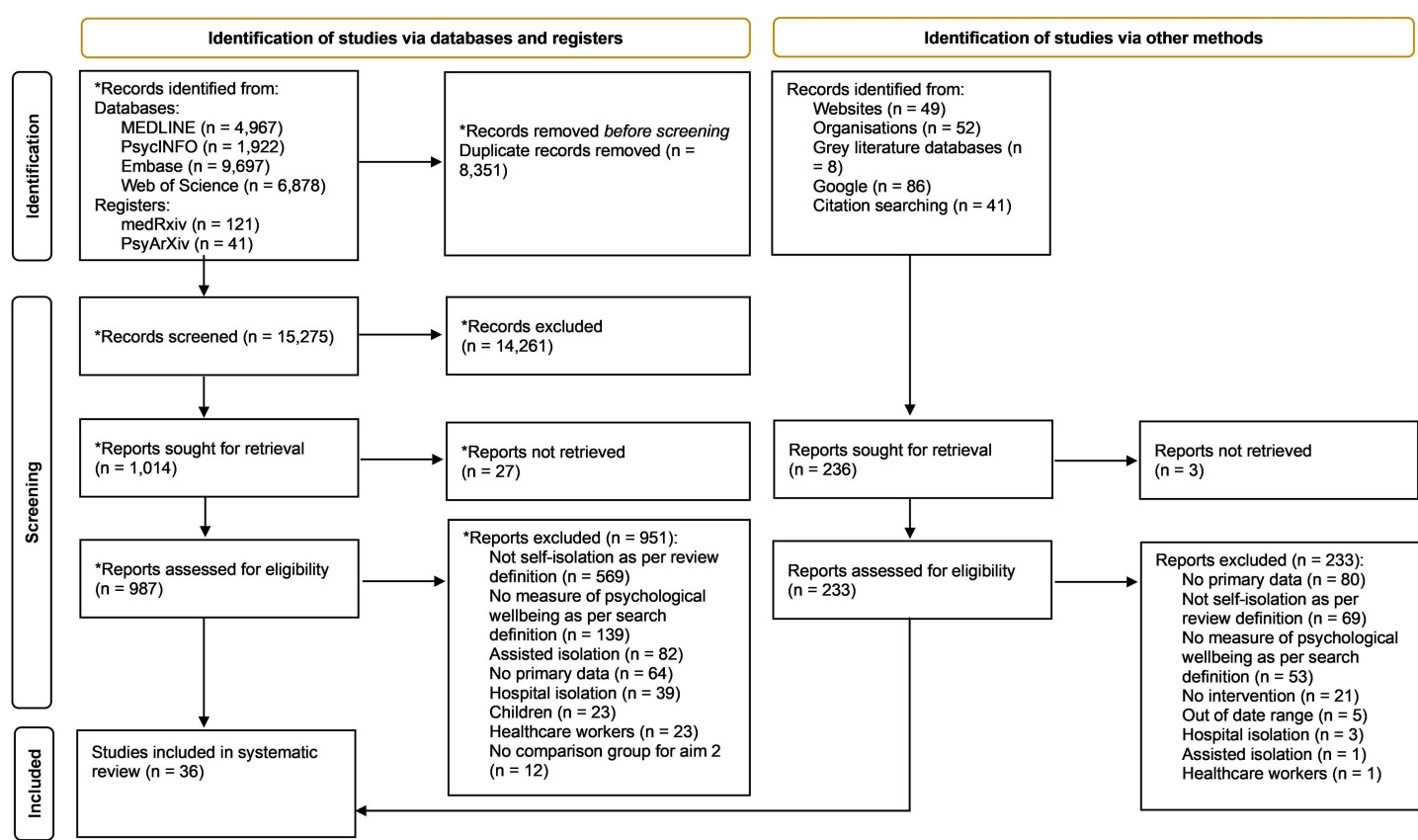

Note. *At this stage, citation screening was completed for this systematic review and a systematic review investigating adherence to self-isolation. Therefore, these totals include citations screened both systematic reviews.

**Fig 1. Study selection flowchart.**

**Table 1. Study characteristics summary.**

| Citation | Region | | | | | | | | Design | | | |
|---|---|---|---|---|---|---|---|---|---|---|---|---|
| | Europe | Asia | South Asia | Africa | East Asia | South America | North America | Multi-continent | Cross-sectional | Longitudinal | Intervention | Qualitative |
| Aaltonen 2022 [19] | | | | | | | | | ✓ | | | |
| Abir 2021 [20] | | | ✓ | | | | | | ✓ | | | |
| Aloba 2021 [21] | | | | ✓ | | | | | ✓ | | | |
| Aslaner 2022 [22] | | ✓ | | | | | | | ✓ | | | |
| Bonsaksen 2020 [23] | ✓ | | | | | | | | ✓ | | | |
| Chakeri 2020 [24] | | ✓ | | | | | | | | | ✓ | |
| Daly 2021 [17] | | | | | | | ✓ | | ✓ | | | |
| Domenghino 2022 [25] | ✓ | | | | | | | | | | | ✓ |
| Flores-Torres 2021 [26] | | | | | | ✓ | | | ✓ | | | |
| Gok 2022 [27] | | ✓ | | | | | | | | | | ✓ |
| Havlioglu 2022 [28] | | ✓ | | | | | | | ✓ | | | |
| Isherwood 2022 [29] | ✓ | | | | | | | | ✓ | | | |
| Jagadeesan 2022 [30] | | | ✓ | | | | | | | | ✓ | |
| Jang 2022 [31] | | | | | ✓ | | | | ✓ | | | |
| Jesmi 2021 [32] | | ✓ | | | | | | | | | | ✓ |
| Joisten 2022 [33] | ✓ | | | | | | | | ✓ | | | |
| Ju 2021 [34] | | | | | ✓ | | | | | ✓ | | |
| Kopilas 2021 [35] | ✓ | | | | | | | | ✓ | | | |
| Kowalski 2021 [36] | ✓ | | | | | | | | ✓ | | | |
| Lohiniva 2021 [37] | ✓ | | | | | | | | | | | ✓ |
| Maric 2022 [38] | ✓ | | | | | | | | ✓ | | | |
| Mohamed 2021 [39] | | | | ✓ | | | | | ✓ | | | |
| Navas 2022 [40] | ✓ | | | | | | | | ✓ | | | |
| Oginni 2021 [41] | | | | ✓ | | | | | ✓ | | | |
| Opakunle, 2022 [42] | | | | ✓ | | | | | ✓ | | | |
| Paz 2020 [43] | | | | | | ✓ | | | ✓ | | | |
| Petrocchi 2021 [44] | ✓ | | | | | | | | ✓ | | | |
| Pheh 2020 [45] | | | ✓ | | | | | | | ✓ | | |
| Plesea-Condratovici 2022 [46] | ✓ | | | | | | | | ✓ | | | |
| Rajagopalan 2022 [47] | | | ✓ | | | | | | | | ✓ | |
| Ripon 2020 [48] | | | ✓ | | | | | | ✓ | | | |
| Rossi 2020 [49] | ✓ | | | | | | | | ✓ | | | |
| Schluter 2022 [50] | | | | | | | | ✓ | ✓ | | | |
| Verberk 2021 [51] | ✓ | | | | | | | | | | | ✓ |
| Wessely 2022 [52] | ✓ | | | | | | | | ✓ | | | |
| Xu 2020 [53] | | | | | ✓ | | | | ✓ | | | |

*Note.*

[*] Among the studies that did not report a lockdown context, only one addressed aim 1 [19], the rest focused on aims 2 and 3.

| Sample size | | | | | Isolation reason | | | | | National/regional lockdown | | | |
|---|---|---|---|---|---|---|---|---|---|---|---|---|---|
| ≤20 | 21-100 | 101-1,000 | 1,001-10,000 | >10,000 | COVID-19 infection | Suspected infection | Close contact | A combination | No reason reported | Lockdown in place | No lockdown in place | Mixed | Not reported** |
| | | ✓ | | | | | | ✓ | | | | | ✓ |
| | | | ✓ | | | | | ✓ | | ✓ | | | |
| | | ✓ | | | ✓ | | | | | | | | ✓ |
| | | ✓ | | | | | | ✓ | | | | | ✓ |
| | | | ✓ | | | | | ✓ | | ✓ | | | |
| | ✓ | | | | ✓ | | | | | | | | ✓ |
| | | ✓ | | | | | | ✓ | | | ✓ | | |
| | | ✓ | | | ✓ | | | | | | | | ✓ |
| | | ✓ | | | | | | ✓ | | ✓ | | | |
| | | ✓ | | | ✓ | | | | | | | | ✓ |
| | | ✓ | | | ✓ | | | | | ✓ | | | |
| | | ✓ | | | | | ✓ | | | | | | ✓ |
| | ✓ | | | | ✓ | | | | | | | | ✓ |
| | | ✓ | | | | | | ✓ | | | | | ✓ |
| ✓ | | | | | ✓ | | | | | | | | ✓ |
| | | | ✓ | | | | | ✓ | | | | | ✓ |
| | | ✓ | | | ✓ | | | | | | | | ✓ |
| | | ✓ | | | | | ✓ | | | | | ✓ | |
| | | ✓ | | | ✓ | | | | | | | | ✓ |
| | ✓ | | | | | | | ✓ | | | | | ✓ |
| | | ✓ | | | | | ✓ | | | | ✓ | | |
| | ✓ | | | | ✓ | | | | | | | | ✓ |
| | ✓ | | | | ✓ | | | | | ✓ | | | |
| | | ✓ | | | | ✓ | | | | ✓ | | | |
| | | ✓ | | | ✓ | | | | | | | | ✓ |
| | | ✓ | | | | | | ✓ | | | | | ✓ |
| | | ✓ | | | ✓ | | | | | ✓ | | | |
| | | ✓ | | | | | | | ✓ | ✓ | | | |
| | | ✓ | | | ✓ | | | | | | | | ✓ |
| | ✓ | | | | ✓ | | | | | | | | ✓ |
| | | ✓ | | | ✓ | | | | | ✓ | | | |
| | | | ✓ | | | | | ✓ | | ✓ | | | |
| | | ✓ | | | | | | ✓ | | | | ✓ | |
| ✓ | | | | | | | ✓ | | | | | | ✓ |
| | | ✓ | | | | | | ✓ | | | | | ✓ |
| | | ✓ | | | | | ✓ | | | | | | ✓ |

**Table 2. Quantitative synthesised results.**

**Panel A. The impact of isolation and quarantine (self-isolation) on psychological symptoms and/or diagnosis**

| | Anxiety | Depressive | General psychological | PTSD | Stress |
|---|---|---|---|---|---|
| Isolating (yes) | ↔[26] ↔[38] ↑[49] ↔[45] ↔[35] ↔[41] ↑[50] | ↑[26] ↔[38] ↔[49] ↔[35] ↔[41] ↑[50] | ↔[19] ↔[26] ↔[38] ↑[20] ↑[45] ↑[17] | ↑[23] ↑[49] | ↔[49] ↔[35] |

**Panel B. Factors associated with psychological symptoms and/or diagnosis during isolation and quarantine (self-isolation)**

| Factor type | Factor | Anxiety | PTSD | Stress | Depressive | General psychological | PTSD | Insomnia |
|---|---|---|---|---|---|---|---|---|
| Isolation | Close contact (vs infected) | ↔[44] | | ↔[19] | ↑[44] | ↔[19] ↑[33] | | |
| | Hotel/hospital (vs home) | ↔[44] ↔[50] | | ↔[39] | ↓[44] ↔[48] ↔[50] ↑[39] | | ↓[48] ↓[39] | |
| | Duration (end versus start of self-isolation) | ↓[34] | | | ↓[34] | | | |
| | Covid-related stressors (yes) | ↑[43] | | | ↑[31] ↑[43] | ↑[36] | | ↑[21] |
| | Assistance/support (no) | | | | ↑[31] | ↑[36] | | |
| Demographic | Age (younger) | | | | ↑[31] | ↓[29] | | |
| | Gender (female) | ↔[41] ↑[43] | | ↑[46] | ↔[31] ↔[41] ↑[43] | ↑[29] | | ↔[21] |
| | Living alone (yes) | | | | ↔[31] | ↔[29] | | |
| Covid-symptoms | Viral load/covid symptoms (higher) | | | ↔[42] | ↔[42] | ↑[36] ↑[42] | | ↔[21] ↔[42] |
| Mental/physical health | Psychological symptoms (any; yes/higher) | | | ↑[53] ↑[46] | | ↑[36] | | ↑[21] |
| | Poor physical health (yes) | | | | ↑[31] | ↑[36] | | |
| | Coping strategies (no) | | | ↑[53] | | ↑[36] | | |

**Panel C. The impact of interventions to improve psychological symptoms and/or diagnosis during isolation and quarantine (self-isolation)**

| Intervention | Anxiety | Depressive | Stress | Insomnia | Adaptive wellbeing |
|---|---|---|---|---|---|
| Telenursing | ↓[24] | | | | |
| Yogic meditation* | ↓[30] ↔[47] | ↓[30] ↓[47] | ↓[30] ↔[47] | ↓[30] ↓[47] | ↑[30] ↑[47] |

*Note.* Panel A = aim 1 the impact of self-isolation on wellbeing, up/down arrow indicates a significant increase/decrease of symptoms in the isolating group, horizontal arrow indicates no effect identified; Panel B = aim 2 factors associated with wellbeing, up/down arrow indicates a significant increase/decrease of symptoms in the group in parenthesis, horizontal arrow indicates no effect identified; Panel C = aim 3 wellbeing interventions during isolation, down arrow indicates a significant decrease of symptoms following the intervention; effect arrows are ordered by risk of bias (risk of bias = ROBINS-E/I [Exposure/Intervention]) white cell colour = some concerns, blue cell colour = high risk, orange cell colour = very high risk'.

characteristics of each study are reported in S5 Appendix. Data were narratively synthesised for each aim using SWiM, resulting in no missing data from the included studies.

## Aim 1: The impact of self-isolation on psychological wellbeing

Associations between self-isolation and wellbeing were reported in 11 quantitative studies, several of which reported more than one wellbeing outcome. The most reported outcomes were anxiety symptoms [26,35,38,41,45,49,50], depressive symptoms [26,35,38,41,49,50], general psychological symptoms [17,19,20,26,45], post-traumatic stress disorder (PTSD) [23,49], and stress related symptoms (Table 2, Panel A) [35,49]. Several other outcomes were reported by only one study, such as loneliness and substance use, which are not synthesised here but can be found in the full extraction tables (S5 Appendix). Evidence was inconsistent; grouping by lockdown context did not alter the pattern of results. Risk of bias is summarised and then the outcomes reported in the highest number of studies are discussed first.

In the risk of bias assessments, no quantitative findings were low risk. Five findings had some concerns [19,23,26,38,49], five were high risk [20,35,41,45,50], and one was very high risk [17]. Only one of the four qualitative studies scored more than 40% on the quality appraisal tool [51], and even the higher-quality study was at substantial risk of bias due to the authors' analysis using an a priori framework based on 'areas of interest'. The risk of bias summaries are reported in S6 Appendix .

For anxiety and depressive symptoms, most studies found no evidence of an effect of self-isolation ([26,35,38,41,45] and [35,38,41,49] respectively) while two reported worse symptoms in those who had self-isolated ([49,50] and [26,50] respectively). Limiting findings to studies at the lowest risk of bias amongst those included (some concerns) [26,38,49] did not change the pattern of findings.

For general psychological symptoms, three studies found no evidence for an association with self-isolation [19,26,38]. Two studies that reported worse general psychological symptoms in those who had self-isolated were carried out under rapidly changing societal contexts [17,20]. Limiting findings to studies at lowest risk of bias amongst those included (some concerns) [19,26,38] suggested no evidence for an effect of self-isolation.

Studies on PTSD symptoms consistently reported a positive association with self-isolation, both were large population cohort studies early in the pandemic, and both had some concerns of bias [23,49]. Two studies found no evidence for an association between self-isolation and with stress, also early in the pandemic, at high risk and some concerns of bias respectively [35,49].

Evidence for an impact of self-isolation on psychological symptoms was more consistent in qualitative studies (Table 3). Participants also described feeling lonely [25,27,32], sad [25,27], angry and frustrated [25,51], bored [27,32,37,51], and afraid [32,51]. They also reported negative impacts on their family relationships [25,37]. In contrast, some also reported self-isolation to be relaxing and providing more time with family [27,37].

## Aim 2: Factors associated with psychological wellbeing during or following self-isolation

Factors associated with wellbeing were reported in 20 quantitative studies, several of which investigated associations with more than one factor (Table 2, Panel B). The most reported factors were related to self-isolation [19,21,31,33,34,36,39,43,44,48,50], demographics [21,29,31,41,43,46], mental and physical health [21,31,36,46,53], and COVID-19 symptoms [21,36,42]. Several factors or outcomes were reported by only one study, such as the time of year or loneliness, which are not reported here but can be found in the full extraction tables (S5 Appendix). Evidence was often inconsistent.

**Table 3. Qualitative synthesised results.**

| Theme | Subthemes |
|---|---|
| Aim 1: Impact on wellbeing | • Perceived increase of depressive and/or anxiety symptoms, stress, or an overall worsening of their mental health [25,51]. |
| | • Negative feelings including loneliness and isolation [25,27,32], sadness [25,27], anger and frustration, including increased aggression [25,51], boredom [27,32,37,51], and fear [32,51].<br>• Some people reported negative impacts on their family relationships [25,37].<br>• Having more time, for example to spend with family and to relax, allowed some to refocus and appreciate what they had [25,37,51]. |
| Aim 2: Factors aggravating poor wellbeing | • People already struggling with mental health before self-isolating felt that it worsened their symptoms [25]. |
| | • Financial difficulties or job insecurity were not reported often, but those who experienced such worries were highly stressed [25,32]. |
| | • Unsupportive managers at work, particularly when managers were perceived to have 'blamed' people for getting sick [25]. |
| | • People with children often reported an impact on their mental health due to conflict between childcare and work priorities [25], or fear of negative impacts on their children, such as a parent dying or inattention whilst parents were unwell [25,27,32,37,51]. |
| | • COVID-19 disease related stressors increased fear, anxiety/panic, and sleep problems. These included not knowing the consequences of infection [25], the experience of COVID-19 symptoms (especially shortness of breath and symptom severity/duration) [32,37], and the wider context (such as knowing someone hospitalised due to COVID-19 and high infection rates and fatalities) [37]. |
| | • Stigma and self-stigma were perceived to be related to worse quality of life, including prolonged voluntary self-isolation [37,51]. |
| | • Excessive media coverage of COVID-19 and conflicting information increased stress and frustration [25,32,37,51]. |
| | • People isolating because they were a close contact of a family member with COVID-19 experienced high emotional burden, because they were often caring for people who were ill, had fears about catching the virus, and did not know how long their self-isolation might go on for [51]. Whereas people isolating because of a positive test worried about escalations of their symptoms, their family's health and felt guilty that they might infect others [37,51]. |
| Aim 2: Factors mitigating poor wellbeing | • Social support, such as WhatsApp, video calls, and online support groups often helped [25,27,32,37,51], although sometimes the positive effects of these were temporary and were not a replacement for in-person contact [25]. |
| | • Supportive managers at work, who were understanding and checked in regularly during self-isolation [25]. |
| | • Coping strategies during isolation included a positive perspective, making plans for after isolation, a regular routine, spirituality, and self-care [27,32,51]. |
| | • People with children sometimes reported their presence as reducing stress and increasing coping [27]. |
| | • People who did not feel their routine changed much during self-isolation were less affected [37]. |
| | • Receiving COVID-19 information from healthcare professionals [27]. |

In risk of bias assessments, no findings were at low risk, one had some concerns [34], and most findings were at high [21,29,31,40,41,43,44,48,50,52] or very high risk of bias [19,22,28,33,36,39,42,46,53] (S6 Appendix). The domains that were high risk overall were: bias due to missing data (79%), confounding (53% of studies), selection of the reported result (26%), the outcome measurement (11%), and participant selection (5%). Limiting findings to some concerns of bias or high risk of bias did not alter the interpretation of the findings for aim 2.

Factors related to self-isolation were reported in eleven studies. Factors included COVID-19 related stressors [31,36,43], the reason for self-isolation [19,33,44], access to support [31,36], duration [21,34], and location [39,44,48,50]. COVID-19 stressors, such as changes in daily life, worries about infection and job security, and a lack of support during self-isolation were consistently associated with higher levels of general psychological, depressive, and anxiety symptoms [31,36,43]. Depressive and anxiety symptoms were found to be lower at the end of self-isolation compared to the start [34], and a longer period of self-isolation was associated with lower sleep problems at the end of the isolation period [21]. The reason for self-isolation

(infection or close contact) was not consistently reported to associate with general psychological symptoms [19,33], or anxiety [19,44], but both studies that examined depression reported that symptoms were higher when people isolated due to infection [19,44]. Self-isolation at home rather than a hotel was associated with lower PTSD symptoms [39,48], but not anxiety symptoms [39,44,50]. Evidence for a relationship between the location of self-isolation and depression was unclear [39,44,48,50], although the outcomes used to measure depression were robust and there were no other obvious differences between the studies that might explain the inconsistent findings.

Factors related to demographics were reported in six studies. Factors included living alone [29,31], age [21,29,31], and gender [29,31,41,43,46]. There was no evidence found for living alone to be associated with wellbeing symptoms [29,31]. There was inconsistent evidence for an association between age or gender with any wellbeing symptoms [21,29,31,41,43,46].

Factors related to mental or physical health were reported in five studies, all of which reported a consistent negative impact on wellbeing outcomes. Psychological disorders and symptoms before or during self-isolation were associated with psychological burden [36], anxiety symptoms [46,53], and insomnia [21]. Poor physical health was associated with psychological burden [36], and depression symptoms [31]. Low levels of coping strategies were associated with psychological burden [36] and anxiety symptoms [53].

Factors related to COVID-19 were reported in three studies, including higher viral load and more severe symptoms [21,36,42]. Both these factors were associated with an increase in general psychological symptoms [36,42]. Whereas no evidence was found for an association with depressive or anxiety symptoms, or insomnia [21,42].

Qualitative findings largely supported the associations identified in the quantitative findings (Table 3). For example, participants perceived that pre-existing mental health problems worsened during self-isolation [25]. Fears around COVID-19 were also prominent and appeared to contribute to poor wellbeing. Participants described concerns about the consequences of infection [25], the experience of COVID-19 symptoms [32,37], fears of death or dependence [32], concerns about workplace consequences and stigma [25], worries about others due to knowledge of high infection rates and fatalities [37], and fears of transmitting the virus to others [25,32,37,51]. These concerns were perceived to lead to anxiety and sleep problems. Other factors which participants perceived to increase the negative wellbeing impacts of self-isolation included fears around COVID-19 [25,32,37,51], financial difficulties [25,32], stigma and self-stigma [37,51], exposure to excessive media coverage of COVID-19 [32], and conflicting guidance on how to isolate and symptom prognosis [25,37,51]. Factors which participants perceived to reduce the negative impact of self-isolation on wellbeing included social support [25,27,32,37,51], coping strategies such as making plans for after isolation, keeping a regular routine, spirituality, and self-care [27,32,51], and receiving regular and reliable information about COVID-19 from healthcare professionals [27]. In addition, some people also reported a positive impact on their wellbeing, such as having more time to spend with family and to relax, which allowed them to refocus and appreciate what they had [25,37,51]. Notably, the studies rated as highest in quality were the only two which reported on stigma [37,51] (S6 Appendix). Concerns about COVID-19 symptoms, fears of transmitting the virus, and exposure to media coverage and conflicting guidance were reported as concerns in both lower- and higher-quality studies. Positive psychological effects were also reported in both lower- and higher-quality studies.

Qualitative findings also indicated that the same factor could have a positive or negative impact on wellbeing, depending on the person's individual context. For example, having children was often reported as a risk factor for a negative impact on wellbeing during isolation, and one context under which this would occur was when there was a perceived conflict

with childcare and work priorities [25]. On the other hand, some people who were worried during their isolation reported that the presence of children reduced stress and helped them to cope [27]. Uncertainties also seemed to play an important role that changed depending on context. For example, for those who were infected, not knowing how bad symptoms might get impacted their wellbeing [25], whereas for those who were isolating due to infection within their household, not knowing how long isolation might go on for impacted their wellbeing [32,37].

### Aim 3: Interventions to improve psychological wellbeing during self-isolation

Three studies investigated the effect of an intervention on wellbeing [24,30,47]. One study tested a telenursing intervention, reporting a greater reduction in anxiety symptoms from pre-test to post-test in the intervention group compared to the control group [24]. Two studies tested yogic meditation interventions, reporting a decrease in depressive and general psychological symptoms and insomnia, and an increase in adaptive wellbeing at the end of the intervention period compared to the start (Table 2, Panel C) [30,47]. Both meditation intervention studies were conducted in the same hospital, using the same outcomes and similar interventions, but at least partially different participants. Study quality was problematic for all interventions, which were rated as high risk [24] or very high risk (S6 Appendix) [30,47].

## Discussion

This systematic review summarises the literature on self-isolation and psychological and emotional wellbeing during the COVID-19 pandemic. Overall, there was considerable heterogeneity in wellbeing outcomes and isolation contexts reported in the studies. Most studies were cross-sectional, survey based, and at high or very high risk of bias. Quantitative evidence for an association between self-isolation and wellbeing was inconsistent, although some clear associations emerged for specific outcomes such as increased PTSD symptoms. Qualitative evidence showed a more consistent negative impact of self-isolation. Stressors, including pre-existing health needs and low levels of support, were consistently associated with worse wellbeing outcomes. Intervention studies were rare and at high or very high risk of bias.

Quantitative studies largely reported psychopathology symptom scales, which may explain the discrepancy between their findings and those of qualitative studies. These scales rarely assessed broader aspects of wellbeing such as worry, stigma, and somatic pain that could lead to behavioural changes like obsessive protective behaviours lasting far beyond the period of self-isolation, as found in other infectious disease contexts [25,37,54,55]. These broader aspects of wellbeing, and others such as frustration and agitation, may also have broader consequences, including lowering trust in governments and reducing adherence to mitigation measures [56,57]. Future research should consider wellbeing beyond psychopathology symptoms to better characterise the impact of self-isolation on wellbeing, whilst continuing to focus on identifying groups with symptom levels suggestive of a treatment need.

One exception was PTSD symptoms, which were consistently higher in people who were currently or previously in self-isolation [23,49]. This is consistent with a previous review that found increased levels of PTSD symptoms in quarantined individuals across all contexts, including different infectious diseases and different groups such as parents, the general population, and healthcare workers [3]. Additionally, home isolation was consistently associated with reduced PTSD symptoms compared to isolation in a managed facility [39,48]. These results suggest that home isolation should be prioritised to reduce the risk of PTSD symptoms during self-isolation.

Self-isolation was found to associate with wellbeing differently, depending on individual factors and context. There was good evidence that people with greater mental and physical health needs [25,28,31,36,43,46,53], who experienced COVID-related stressors including inadequate support [25,31,32,36,37,43,51], and had reduced coping strategies [36,52,53], were most at risk of adverse outcomes. Of interest, the two studies that examined viral load reported an increase in general psychological symptoms (but not other symptoms) during self-isolation [36,42]. Previous research found that viral load is associated with disease severity and duration, particularly in older adults, suggesting it may present a useful biomarker of risk during infection mitigation strategies, warranting further investigation [58].

As found during COVID-19 lockdowns [59,60], parents and carers were more at risk of poor mental health during self-isolation, especially when there was conflict between childcare and work [25]. However, the presence of children could also reduce stress and helped some parents to cope [27]. This complexity mirrors the intricacies observed in prior studies, which found that factors such as the number and age of the children could exert different risk or protective effects [60,61]. Future research should focus on identifying the subpopulations most at risk of adverse wellbeing outcomes during self-isolation and developing and evaluating targeted public health interventions to support these groups, including providing practical support and promoting coping strategies to those who need it.

Few studies investigating the effect of interventions on wellbeing during self-isolation were identified. All three studies reported a supposed impact of the intervention on most wellbeing outcomes [24,30,47], but several non-intervention studies also found that people generally experienced a reduction of symptoms over the course of isolation [21,34]. This suggests that two of the intervention studies, which lacked control groups, may have overestimated the effect of the intervention. Nevertheless, the study that used a control group found a greater reduction in anxiety at the end of the intervention (i.e., at follow-up) in those who received daily tele-nursing [24]. Qualitative studies highlighted modifiable stressors that have been consistently reported in previous infectious disease and disaster contexts, such as inadequate and conflicting information [54,62,63], leading to heightened fears of disease progression and extended isolation [64–66]. Together, these findings suggest that high-quality intervention studies should be prioritised to better understand how to mitigate the impact of self-isolation on wellbeing. Emphasis should be placed on interventions targeting loneliness worries, and misinformation, for example, through regular contact with healthcare professionals, whilst monitoring and identifying individuals who may require additional support.

### Strengths and limitations

To our knowledge, this is the first review to explore the relationship between self-isolation and psychological wellbeing during the COVID-19 pandemic. Strengths include a pre-registered protocol, no geographical or language limitations, comprehensive risk of bias assessments, a robust process for agreement, and adherence to PRISMA, SWiM, and eMERGe guidelines. Limitations of the studies were the inconsistent use of the terms 'isolation' and 'quarantine', which could have led to the exclusion of relevant research, and the high risk of bias in many studies. Limitations of the review were first, seeing as we synthesised results from studies conducted around the globe, the sample included in individual studies may be representative of the general population of that country, but may not be representative of other countries. Second, we were unable to formally analyse publication bias and did not examine heterogeneity in study design in the synthesis, which limits our confidence in the certainty of the evidence presented. Human error may have resulted in missed studies.

## Conclusions

Self-isolation can impact psychological wellbeing, especially PTSD symptoms, but quantitative data shows mixed results for other wellbeing outcomes. Self-isolating at home may reduce the risk of PTSD symptoms compared to a managed facility such as a hotel, but more and better-quality evidence is needed across all wellbeing outcomes. When implementing self-isolation directives in the future, public health officials should make it a priority to support individuals with pre-existing physical and mental health conditions, a lack of support, or those who face additional life stressors. Clinicians and healthcare workers can play a key role in identifying and supporting those most at risk. Interventions should focus on addressing loneliness, worries, and misinformation, improving coping strategies, and monitoring and identifying individuals who need additional support.

## Supporting information

**S1 Appendix  PRISMA checklists.**
(PDF)

**S2 Appendix.  Study protocol and deviations.**
(PDF)

**S3 Appendix.  Searches.**
(PDF)

**S4 Appendix.  Full text exclusion reasons.**
(PDF)

**S5 Appendix.  Data extraction.**
(PDF)

**S6 Appendix.  Study quality.**
(PDF)

## Author contributions

**Conceptualization:** Alex F. Martin, Louise E. Smith, Richard Amlôt, Gideon James Rubin.

**Data curation:** Alex F. Martin.

**Formal analysis:** Alex F. Martin, Louise E. Smith, Samantha K. Brooks.

**Investigation:** Alex F. Martin, Louise E. Smith, Samantha K. Brooks, Madeline V. Stein, Rachel Davies, Neil Greenberg, Gideon James Rubin.

**Methodology:** Alex F. Martin, Louise E. Smith, Samantha K. Brooks, Neil Greenberg, Gideon James Rubin.

**Project administration:** Alex F. Martin.

**Supervision:** Louise E. Smith, Richard Amlôt, Gideon James Rubin.

**Validation:** Alex F. Martin, Samantha K. Brooks, Madeline V. Stein, Rachel Davies, Richard Amlôt, Gideon James Rubin.

**Visualization:** Alex F. Martin.

**Writing – original draft:** Alex F. Martin.

**Writing – review & editing:** Alex F. Martin, Louise E. Smith, Samantha K. Brooks, Madeline V. Stein, Rachel Davies, Richard Amlôt, Neil Greenberg, Gideon James Rubin.

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
