## [Decision Letter · Decision Letter 0]

2 Jul 2024

PONE-D-24-15958The impact of self-isolation on psychological wellbeing and how to reduce it: a systematic reviewPLOS ONE

Dear Dr. Martin,

Thank you for submitting your manuscript to PLOS ONE. After careful consideration, we feel that it has merit but does not fully meet PLOS ONE’s publication criteria as it currently stands. Therefore, we invite you to submit a revised version of the manuscript that addresses the points raised during the review process.

We look forward to receiving your revised manuscript.

Kind regards,

AKM Alamgir, PhD

Academic Editor

PLOS ONE

[This work was carried out at King’s College London. LES, RA, and GJR were participants of the UK’s Scientific Advisory Group for Emergencies or its subgroups. GJR advised the UK’s Office for National Statistics on its work relating self-isolation – papers relating to this work were considered as part of the review. All authors co-authored papers that were considered during the review process. RA is an employee of the UK Health Security Agency. AFM, SKB, RD, MVS, and NG report no competing interests.]. 

4.  review your reference list to ensure that it is complete and correct. If you have cited papers that have been retracted, please include the rationale for doing so in the manuscript text, or remove these references and replace them with relevant current references. Any changes to the reference list should be mentioned in the rebuttal letter that accompanies your revised manuscript. If you need to cite a retracted article, indicate the article’s retracted status in the References list and also include a citation and full reference for the retraction notice.

Additional Editor Comments (if provided):

Reviewers' comments:

Reviewer's Responses to Questions

**Comments to the Author**

1. Is the manuscript technically sound, and do the data support the conclusions?

Reviewer #1: Partly

Reviewer #2: Yes

2. Has the statistical analysis been performed appropriately and rigorously? 

Reviewer #1: I Don't Know

Reviewer #2: Yes

3. Have the authors made all data underlying the findings in their manuscript fully available?

Reviewer #1: Yes

Reviewer #2: Yes

4. Is the manuscript presented in an intelligible fashion and written in standard English?

Reviewer #1: Yes

Reviewer #2: Yes

5. Review Comments to the Author

Reviewer #1: This paper examines how self-isolation after contracting or being in close contact with someone who contracted COVID-19 impacted psychological wellbeing, as well as what factors exacerbated or mitigated self-isolation’s impacts on wellbeing. Through a systematic review, the authors put aside questions about lockdown’s impacts in favour of investigating the experiences of those suspected or known to carry the virus — an increasingly relevant question as public health measures end but isolation protocols become the norm.

In their paper, the authors conduct a systematic review of qualitative and quantitative studies tracking psychological outcomes for adults self-isolating at home, with some comparison to those isolating in a hospital or a hotel. The included studies examined a wide variety of psychological outcomes, namely anxiety, depression, general wellbeing, PTSD, and stress. The authors examined various factors that could alter self-isolation’s impacts including age, gender, and pre-existing mental health issues. The inclusion of both qualitative and quantitative studies allowed the authors to examine a wide variety of risk factors and outcomes, with varying levels of support for relationships between these factors and outcomes.

The authors found that, despite wide agreement among qualitative studies that self isolation corresponded to worse psychological wellbeing, quantitative studies appear more mixed on social isolation’s impacts. Specifically, isolation corresponds to worsened PTSD symptoms but did not consistently correspond to increased anxiety, depression, stress, or general psychological issues. The groups most at risk for reduced wellbeing associated with isolation appear to be those infected with the virus, people with longer isolation periods, people without assistance, and those with pre-existing mental health conditions. Some factors, such as having children, had inconsistent impacts. Overall, many of the studies examining interventions or outcomes had concerns of bias.

The main strengths of this paper are its rigorous design, wide scope of literature review, and the steps taken to reduce and accurately report bias both between reviewers and within the papers themselves. It also answers a timely question relevant to continuing inquiry into the pandemic’s impacts and ways to mitigate the psychological impacts of future pandemics. One weakness was inconsistencies between the stated conclusions and the paper’s findings. For instance, the conclusion states that self-isolation impacts psychological wellbeing, but the quantitative data appear far more mixed on isolation’s actual effects outside of its impacts on PTSD.

Some more minor issues to address:

1. The authors may want to justify only searching UK agencies and organisations, as this may introduce bias toward sources in the UK context

2. The authors should clarify earlier in the study that the review only considered adults isolating at home, since this limits the review’s applicability

3. The authors note that the groupings were not defined a priori (line 159-162). The authors may want to specify how these groupings were chosen

4. The description of study characteristics would flow better if the authors resynthesized the information in a table, which would be more helpful in assessing bias

5. The methods section notes that the review included adults who isolated at home and excluded children, healthcare workers, and those in managed facilities. However, some of the included studies do not explicitly have age as an inclusion criteria listed in table 1 (for example, Chakeri et al., 2020 and Plesea-Condratovici et al., 2022). The authors should either update the table to reflect whether the data only included adults or otherwise justify those studies’ inclusion

6. Table 2, specifically panel B, is somewhat difficult to follow. On first reading, it was not evident whether the arrows indicated difference in effect sizes comparing the groups mentioned to some control or the observed effects among that specific group. In addition, the colours in table 2 differed from that presented in the note. The authors should specify what the arrows entail by either updating the panel’s title or altering the note.

7. Despite a consistent association between viral load and an increase in psych symptoms, the authors do not address this further.

8. The decision to focus on adults as opposed to young people is a large deviation from the registered protocol that may warrant mentioning in the body of the paper.

Reviewer #2: The manuscript is technically sound, with data supporting the conclusions. The statistical analysis has been performed appropriately and rigorously, utilizing tools like the Risk of Bias in Non-randomized Studies for Exposure (ROBINS-E) and Interventions (ROBINS-I) to ensure the validity of findings. The authors have made all underlying data fully available, providing detailed explanations of extracted data, study characteristics, and synthesis methods. The manuscript is presented in an intelligible fashion and written in standard English, though some areas could benefit from improved clarity and readability.

Colour Blindness Accessibility in Table 2 - I suggest using an intuitive yet accessible color scheme for Table 2 to accommodate readers with color blindness. This ensures that all readers can interpret the data effectively.

Demographic Characteristics - Including a short discussion on the demographic characteristics of all study participants would help identify who these results may be applicable to or not. This consideration is important for understanding the generalizability of the findings.

Exclusion of Social Isolation - The authors do not explain why social isolation is excluded from the scope of the study. Since social isolation and self-isolation could overlap, both involving reduced social interactions, providing a rationale for this exclusion would enhance the manuscript's clarity.

Clarification of "Wellbeing" - "wellbeing" is broad. Clarifying which aspects of wellbeing are being examined would be beneficial. For example, specifying "The impact of self-isolation during the COVID-19 pandemic on psychological and emotional wellbeing" would provide clearer context.

Definition of Self-Isolation - The definition of self-isolation could be clearer. The current definition mixes the concepts of isolation and quarantine. Consider rephrasing for better clarity: "Self-isolation in this study is defined as both isolation (separating those who are sick from those who are well) and quarantine (separating those at risk of illness from those who are well)."

Readability:

The rationale in the introduction "How self-isolation specifically impacted wellbeing is less clear” could be reworded to improve clarity and enhance the overall comprehension of the manuscript.

Overall, the manuscript is robust and well-presented, but these suggestions aim to improve accessibility, clarity, and completeness, thereby enhancing the manuscript’s impact and readability.

6. PLOS authors have the option to publish the peer review history of their article (what does this mean? ). If published, this will include your full peer review and any attached files.

**Do you want your identity to be public for this peer review?** For information about this choice, including consent withdrawal, please see our Privacy Policy .

Reviewer #1: No

Reviewer #2: **Yes: ** Bay Bahri

---

## [Author Response · Author response to Decision Letter 1]

22 Aug 2024

We are grateful to the reviewers for their feedback and insightful comments that have considerably improved the manuscript, particularly around clarity and consistency, accessibility, and readability. Please find our responses below.

Reviewer 1

1. One weakness was inconsistencies between the stated conclusions and the paper’s findings. For instance, the conclusion states that self-isolation impacts psychological wellbeing, but the quantitative data appear far more mixed on isolation’s actual effects outside of its impacts on PTSD.

Thank you for this important point, we completely agree. We have revised the text accordingly in the conclusion on p22:

‘Self-isolation can impact psychological wellbeing, especially PTSD symptoms, but quantitative data shows mixed results for other wellbeing outcomes. Self-isolating at home may reduce the risk of PTSD symptoms, but more and better-quality evidence is needed across all wellbeing outcomes. When implementing self-isolation directives in the future, public health officials should make it a priority to support vulnerable individuals with pre-existing physical and mental health conditions, a lack of support, or those who face additional life stressors. Clinicians and healthcare workers can play a key role in identifying and supporting those most at risk. Interventions should focus on addressing loneliness, worries, and misinformation, improving coping strategies, and monitoring and identifying individuals who need additional support.’

Some more minor issues to address:

2. The authors may want to justify only searching UK agencies and organisations, as this may introduce bias toward sources in the UK context

Please find the method updated accordingly on pp.4-5:

‘A Google search and searches of the websites for relevant UK agencies and organisations (e.g. the UK Testing Initiatives Evaluation Board, the UK Office for National Statistics), and direct inquiries with UK Government agencies were used to identify other potentially relevant sources. We searched only UK agencies due to our team’s experience. Combined with the UK-based Google search, this may have resulted in missing some grey literature from outside the UK (although we note that these searches did not find any studies for inclusion).’

3. The authors should clarify earlier in the study that the review only considered adults isolating at home, since this limits the review’s applicability

We agree and have updated the manuscript in several places to reflect this:

The title on p.1: ‘The impact of self-isolation on psychological wellbeing in adults and how to reduce it: a systematic review’

The abstract on p.2: ‘Objective: To synthesise evidence on the impact of self-isolation at home on the psychological wellbeing of adults in the general population during the COVID-19 pandemic.’

The introduction on p.3: ‘Most studies during the COVID-19 pandemic focused on the impact of "lockdown" measures (broad population stay-at-home orders) on psychological and emotional* wellbeing (hereafter wellbeing) [2]. However, the impact of home-based self-isolation on adult wellbeing globally has not yet been systematically reviewed.**

This is important given that a) studies carried out in other infectious disease contexts indicate that self-isolation may be associated with psychopathology symptoms and broader wellbeing outcomes in adults, such as insomnia and substance use.’

*Emotional wellbeing was added in response to reviewer 2, point 4.

** The last sentence has been amended in response to reviewer 2, point 6.

4. The authors note that the groupings were not defined a priori (line 159-162). The authors may want to specify how these groupings were chosen

Thank you, we have updated the manuscript to specify how groups were chosen on p.8:

‘For aim 1, we compared outcomes for those in self-isolation with those not self-isolating. For aim 2, we also grouped studies by factor (isolation, demographic, COVID-19, and mental/physical health characteristics) and associations between factors and self-isolation were reported. These groupings were not defined a priori, but the themes emerged during the process of extracting the factors and were revised iteratively as we synthesised the findings.’

5. The description of study characteristics would flow better if the authors resynthesized the information in a table, which would be more helpful in assessing bias

We have now summarised the study characteristics in Table 1 (pp10-11) and moved the full version of the table to the appendix (S4.1 Table).

6. The methods section notes that the review included adults who isolated at home and excluded children, healthcare workers, and those in managed facilities. However, some of the included studies do not explicitly have age as an inclusion criteria listed in table 1 (for example, Chakeri et al., 2020 and Plesea-Condratovici et al., 2022). The authors should either update the table to reflect whether the data only included adults or otherwise justify those studies’ inclusion

This is a very good point, thank you. We have updated Table 1 – now appendix (S4.1 Table) - to show why studies were included if age was not specified in the study inclusion criteria (5 studies).

7. Table 2, specifically panel B, is somewhat difficult to follow. On first reading, it was not evident whether the arrows indicated difference in effect sizes comparing the groups mentioned to some control or the observed effects among that specific group. In addition, the colours in table 2 differed from that presented in the note. The authors should specify what the arrows entail by either updating the panel’s title or altering the note.

We have updated the note for Table 2 (p.13) so that it shows more clearly what the arrow direction indicates in each panel. As suggested by reviewer 2 (point 1), we have also updated the colour scheme so that it is accessible for those with colour-blindness:

‘Note. Panel A = aim 1 the impact of self-isolation on wellbeing, up/down arrow indicates a significant increase/decrease of symptoms in the isolating group, horizontal arrow indicates no effect identified; Panel B = aim 2 factors associated with wellbeing, up/down arrow indicates a significant increase/decrease of symptoms in the group in parenthesis, horizontal arrow indicates no effect identified; Panel C = aim 3 wellbeing interventions during isolation, down arrow indicates a significant decrease of symptoms following the intervention; effect arrows are ordered by risk of bias (risk of bias = ROBINS-E/I [Exposure/Intervention]) dark blue colour = some concerns, red colour = high risk, black colour = very high risk’

Reference: The best charts for color blind viewers | Blog | Datylon

8. Despite a consistent association between viral load and an increase in psych symptoms, the authors do not address this further.

This is an interesting point and viral load is already included in the results. We have now added the following to the discussion on p.20:

Self-isolation was found to associate with wellbeing differently, depending on individual factors and context. There was good evidence that people with greater mental and physical health needs [24, 27, 30, 35, 42, 45, 52], who experienced COVID-related stressors including inadequate support [24, 30, 31, 35, 36, 42, 50], and had reduced coping strategies [35, 51, 52], were most at risk of adverse outcomes. Of interest, the two studies that examined viral load reported an increase in general psychological symptoms (but not other symptoms) during self-isolation [35, 41]. Previous research found that viral load is associated with disease severity and duration, particularly in older adults, suggesting it may present a useful biomarker of risk during infection mitigation strategies, warranting further investigation [57].

9. The decision to focus on adults as opposed to young people is a large deviation from the registered protocol that may warrant mentioning in the body of the paper.

We have now included this in the method on p.4:

The protocol was prospectively registered on PROSPERO (CRD42022378140). The biggest change from the protocol was to only include studies of adults in this review, due to the number of studies identified in the initial search, deviations from the protocol are reported in full in S2 Appendix.

Reviewer 2

1. Colour Blindness Accessibility in Table 2 - I suggest using an intuitive yet accessible color scheme for Table 2 to accommodate readers with color blindness. This ensures that all readers can interpret the data effectively.

Thank you for this important point. After considering several recommendations for accessibility, we have changed the arrows in Table 2 (p.13) to contrasting colours, using dark blue, red, and black. Reference: The best charts for color blind viewers | Blog | Datylon

2. Demographic Characteristics - Including a short discussion on the demographic characteristics of all study participants would help identify who these results may be applicable to or not. This consideration is important for understanding the generalizability of the findings.

We have discussed the point, and it is not clear how we can present the demographic characteristics of study participants, because a) we are not pooling all studies as in a meta-analysis, and b) different countries will have different representative socio-demographics. We have updated the text to make this limitation clearer:

Results (p.9): ‘Studies investigated the general population, rather than sub-groups within the population.’

Limitations (p.22): ‘Limitations of the review were first, seeing as we synthesised results from studies conducted around the globe, the sample included in individual studies may be representative of the general population of that country, but may not be representative of other countries. Second, we were unable to formally analyse publication bias…’

3. Exclusion of Social Isolation - The authors do not explain why social isolation is excluded from the scope of the study. Since social isolation and self-isolation could overlap, both involving reduced social interactions, providing a rationale for this exclusion would enhance the manuscript's clarity.

Social isolation was excluded from the search parameter related to the identification of self-isolation, because it affected the integrity of this element of the search (i.e. to capture studies on self-isolation as per our study definition). We have updated the text to make this point clearer in the method, p.5:

‘The search strategy included terms for COVID-19 AND isolation and quarantine (combined with NOT social isolation) AND psychological wellbeing. Social isolation was excluded as a self-isolation search term because it generated a large number of citations that did not meet our definition of isolation or quarantine.’

4. Clarification of "Wellbeing" - "wellbeing" is broad. Clarifying which aspects of wellbeing are being examined would be beneficial. For example, specifying "The impact of self-isolation during the COVID-19 pandemic on psychological and emotional wellbeing" would provide clearer context.

Thank you, we have added this clarification where we provide:

• the abstract (p.2) Objective: To synthesise evidence on the impact of self-isolation at home on the psychological and emotional wellbeing of adults in the general population during the COVID-19 pandemic.

• the definition of self-isolation in the introduction (p.3) ‘Most studies during the COVID-19 pandemic focused on the impact of "lockdown" measures (broad population stay-at-home orders) on psychological and emotional wellbeing (hereafter wellbeing).’

• the discussion (p.19) ‘This systematic review summarises the literature on self-isolation and psychological and emotional wellbeing during the COVID-19 pandemic.’

5. Definition of Self-Isolation - The definition of self-isolation could be clearer. The current definition mixes the concepts of isolation and quarantine. Consider rephrasing for better clarity: "Self-isolation in this study is defined as both isolation (separating those who are sick from those who are well) and quarantine (separating those at risk of illness from those who are well)."

We have updated the text accordingly on p.3:

‘Self-isolation in this study is defined as both isolation (separating those who are sick from those who are well) and quarantine (separating those at risk of illness from those who are well).’

6. The rationale in the introduction "How self-isolation specifically impacted wellbeing is less clear” could be reworded to improve clarity and enhance the overall comprehension of the manuscript.

Thank you for highlighting this important point of clarification. The text on p3. Has been updated as follows:

‘However, the impact of home-based self-isolation on adult wellbeing globally has not yet been systematically reviewed.’

---

## [Decision Letter · Decision Letter 1]

8 Sep 2024

The impact of self-isolation on psychological wellbeing in adults and how to reduce it: a systematic review

PONE-D-24-15958R1

Dear Dr. Alex Martin,

We’re pleased to inform you that your manuscript has been judged scientifically suitable for publication and will be formally accepted for publication once it meets all outstanding technical requirements.

Kind regards,

AKM Alamgir, PhD

Academic Editor

PLOS ONE

Additional Editor Comments (optional):

Reviewers' comments:

Reviewer's Responses to Questions

**Comments to the Author**

1. If the authors have adequately addressed your comments raised in a previous round of review and you feel that this manuscript is now acceptable for publication, you may indicate that here to bypass the “Comments to the Author” section, enter your conflict of interest statement in the “Confidential to Editor” section, and submit your "Accept" recommendation.

Reviewer #1: (No Response)

Reviewer #2: All comments have been addressed

2. Is the manuscript technically sound, and do the data support the conclusions?

Reviewer #1: Yes

Reviewer #2: (No Response)

3. Has the statistical analysis been performed appropriately and rigorously? 

Reviewer #1: Yes

Reviewer #2: (No Response)

4. Have the authors made all data underlying the findings in their manuscript fully available?

Reviewer #1: Yes

Reviewer #2: (No Response)

5. Is the manuscript presented in an intelligible fashion and written in standard English?

Reviewer #1: Yes

Reviewer #2: (No Response)

6. Review Comments to the Author

Reviewer #1: A few very minor grammatical errors

1. In line 280, the authors begin a sentence with whereas

2. In line 400, the authors left a double question mark that should be replaced with something along the lines of "those isolating in a hotel or hospital"

Otherwise, the authors have addressed all of my concerns.

Reviewer #2: (No Response)

7. PLOS authors have the option to publish the peer review history of their article (what does this mean? ). If published, this will include your full peer review and any attached files.

**Do you want your identity to be public for this peer review?** For information about this choice, including consent withdrawal, please see our Privacy Policy .

Reviewer #1: No

Reviewer #2: **Yes: ** Bay Bahri

---

## [Editor Report · Acceptance letter]

PONE-D-24-15958R1

PLOS ONE

Dear Dr. Martin,

I'm pleased to inform you that your manuscript has been deemed suitable for publication in PLOS ONE. Congratulations! Your manuscript is now being handed over to our production team.

Kind regards,

on behalf of

Dr AKM Alamgir

Academic Editor

PLOS ONE